# Transcriptional Regulation during Aberrant Activation of NF-κB Signalling in Cancer

**DOI:** 10.3390/cells12050788

**Published:** 2023-03-02

**Authors:** Kamalakshi Deka, Yinghui Li

**Affiliations:** 1School of Biological Sciences (SBS), Nanyang Technological University (NTU), 60 Nanyang Drive, Singapore 637551, Singapore; 2Institute of Molecular and Cell Biology (IMCB), A*STAR, Singapore 138673, Singapore

**Keywords:** NF-κB signalling, cancer, chromatin landscape, epigenetic

## Abstract

The NF-κB signalling pathway is a major signalling cascade involved in the regulation of inflammation and innate immunity. It is also increasingly recognised as a crucial player in many steps of cancer initiation and progression. The five members of the NF-κB family of transcription factors are activated through two major signalling pathways, the canonical and non-canonical pathways. The canonical NF-κB pathway is prevalently activated in various human malignancies as well as inflammation-related disease conditions. Meanwhile, the significance of non-canonical NF-κB pathway in disease pathogenesis is also increasingly recognized in recent studies. In this review, we discuss the double-edged role of the NF-κB pathway in inflammation and cancer, which depends on the severity and extent of the inflammatory response. We also discuss the intrinsic factors, including selected driver mutations, and extrinsic factors, such as tumour microenvironment and epigenetic modifiers, driving aberrant activation of NF-κB in multiple cancer types. We further provide insights into the importance of the interaction of NF-κB pathway components with various macromolecules to its role in transcriptional regulation in cancer. Finally, we provide a perspective on the potential role of aberrant NF-κB activation in altering the chromatin landscape to support oncogenic development.

## 1. Introduction

The nuclear transcription factor NF-κB was discovered in 1986 as a **N**uclear **F**actor that binds an immunoglobulin **k**appa light chain of activated **B**-cells [1]. NF-κB was subsequently reported to regulate the expression of various important target genes having diverse physiological functions in multiple cell types through its specific DNA binding activity [2,3]. The family of NF-κB transcription factors in mammals comprises five members—RelA (p65), RelB, c-Rel, NF-κB1 (p105/p50) and NF-κB2 (p100/p52) [4,5,6,7,8,9,10,11,12,13,14,15,16,17,18,19,20,21]. Activation of NF-κB occurs via two major signalling pathways—the canonical and non-canonical pathways that involve distinct regulatory mechanisms and NF-κB members. One of the prominent features of NF-κB transcription factors is their association with the member protein of IκB inhibitor family in the cytoplasm making them unavailable for transcriptional activation in the nucleus. The IκB family typically consists of five members (IκBα, IκBβ, IκBε, IκBζ and BCL3), all sharing similar structures. However, unprocessed p100 and p105 proteins are also categorized as members of the IκB family of proteins due to the presence of typical ankyrin repeats (ANK) in their C-terminal region. An alternative transcript of *p105* gene, only reported to be expressed in some murine lymphoid cells, has also been named as one of the members of the IκB family (IκBγ) [22,23,24]. Hence, activation of both the canonical and non-canonical NF-κB pathway involves phosphorylation-dependent degradation of IκB factors by stimulus-response-activated IκB kinases (IKKs). The canonical NF-κB pathway is mediated through the activation of NF-κB essential modifier (NEMO)-dependent IKK (IKKγ), whereas non-canonical NF-κB pathway activation requires a NEMO-independent kinase complex involving IκB kinase α (IKKα) and the NF-κB-inducing kinase (NIK) [25]. Upon activation of the canonical NF-κB pathway, IκB kinases (IKKα, IKKβ and NEMO) phosphorylates inhibitory IκBs and target the latter for proteasomal degradation, resulting in the subsequent nuclear accumulation of NF-κB dimers [26,27,28,29]. In the non-canonical NF-κB pathway, NIK phosphorylates IKKα on Ser 176 position, which in turn phosphorylates p100 subunit, leading to cleavage and ubiquitination mediated degradation of C-terminal half of p100 protein generating active p52 subunit [30]. However, reports also suggest the presence of atypical nuclear-localized IκB proteins, referred to as the BCL3 subfamily (Bcl3, IκBNS, IκBζ and IκBη). These IκBs are reported to show entirely different sub-cellular localization, activation kinetics and functional diversity. They are not only capable of interacting with NF-κB transcription factors inside the nucleus but are also found to get induced and not degraded after NF-κB activation, compared to typical IκB members. In addition, they do not exclusively act as inhibitors of the NF-κB pathway, instead they can regulate the transcriptional activity of NF-κB transcription factors both positively and negatively [25,31,32,33,34]. Nuclear-localized BCL3 act as transcriptional coactivators by removing suppressive p50/p50 homodimers from the promoter of its target genes, in turn allowing binding of activating p50/p65 heterodimer [35]. Bcl3 is also reported to suppress transcription via blocking of the ubiquitination of p50 to stabilize a suppressive NF-κB complex in the nucleus [36]. One interesting finding on the nuclear role of IκB family proteins is its direct binding to NF-κB target sites. Wang et al., showed that Bcl3 forms a complex with p52 homodimer to activate transcription when bound to G/C-rich κB sites in the DNA, whereas the same complex represses transcription when bound to A/T-centric sites in the DNA [37]. In a recent finding, it is reported that atypical IκB Bcl3 enhances the generation of p52 homodimer, subsequently upregulating the expression of target genes involved in proliferation, migration and inflammation [38]. Another BCL3 family member protein, IκBζ, is reported to inhibit transactivation of p65 and its DNA-binding activity in the nucleus [39]. In contrast, IκBη have been shown to be a positive regulator of NF-κB-mediated expression of pro-inflammatory cytokines [40]. Nuclear IκBNS is also shown to interact with several different NF-κB factors in the nucleus but its biological role towards the activity of NF-κB transcription factors is yet to be elucidated [41,42].

In the presence of activating stimuli, the NF-κB-signalling cascade can be induced via either of the pathways depending on the type of stimuli, dimers formed and kinases involved in the post transcriptional modification (PTM) of IκBs and processing of NF-κB factors. In addition to innate and adaptive immune response-dependent activation of the NF-κB pathway, the range of stimuli activating either the canonical or non-canonical pathway varies to a large extent. The canonical pathway of NF-κB is highly inducible and is activated by a diverse range of stimuli, such as radiation, DNA damage, cytokines (TNF-a, IL-1, IL-6), chemokines (MCP-1, IL-8), growth factors, adhesion molecules (ICAM-1, VCAM-1, ELAM), reactive oxygen species (ROS), pattern-recognition receptors (PRRs) and pro-inflammatory receptors such as TNF receptor superfamily (TNFRs) and Toll Like receptor superfamily (TLRs) [43,44,45,46,47,48,49,50,51]. In contrast, the non-canonical NF-κB pathway relies on specific sets of cytokine/receptor molecules for its activation, such as tumour necrosis factor (TNF) receptor superfamily proteins, including BAFF receptor (BAFFR), CD-40, lymphotoxin β receptor (LTβR), Fn14 and receptor activator of nuclear factor kappa-B (RANK) [52,53,54,55], all of which signal through a MAP3K member kinase (MAP3K14) called NF-κB-inducing kinase (NIK), making it a master regulator of the non-canonical NF-κB pathway [56,57,58,59].

Once activated, each subunit of NF-κB signalling cascade, p65 (RelA), RelB, c-Rel, p105/p50 (NF-κB1) and p100/52 (NF-κB2) associate with each other to form distinct transcriptionally active homo/heterodimers [60]. Though they all possess a conserved 300-amino-acid-long amino-terminal Rel homology domain (RHD) that is important for dimerization, DNA binding and interaction with IκBs, as well as nuclear translocation, the role of transactivation is characterised to specific members. RelA (p65), RelB and c-Rel contain the carboxy-terminal transactivation domains (TAD), which form transcriptionally active heterodimers only with p50 and p52 subunits, in turn assisting in DNA-binding activity and activated target gene expression. [61,62]. Reports also suggest formation of homodimers within Rel proteins as well as p50 and p52 subunits [63,64,65,66]. Gourisankar and his group have solved the crystal structures of homodimers of several NF-κB pathway factors such as p65 (RelA) homodimer in complex with a DNA target (2.4 Å resolution) and p50 homodimer bound to a palindromic κB site (2.3 Å resolution) [12,65]. Interestingly, p50/p50 homodimer has been described to exert inhibitory effects on NF-ĸB regulated gene expression [67,68]. c-Rel homodimer is also reported to have the ability to bind IκBα, which in turn inhibits its DNA binding but not cytoplasmic retention [69,70]. A recent report also showed atypical IĸB protein, Bcl3-mediated enhanced generation of p52 homodimer, in turn enhancing transcription of genes involved in cancer-associated biological processes [38]. There are also many other combinations of dimers reported but the most prominent dimers are the RelA-p50 and RelB-p52 dimers, which are activated by the respective canonical and non-canonical pathways [71,72]. The complexity of the NF-κB-signalling mechanisms is further illustrated by the specificity of NF-κB dimers in the transcriptional activation of different target genes.

In addition, NF-κB member proteins also undergo various post-translational modifications (PTMs), like phosphorylation and acetylation, regulating their interaction and crosstalk with components of other signalling pathways. As mentioned earlier, the phosphorylation status of the IκBs determines the activation state of the NF-κB pathway. The availability and activity of NIK, one of the major activating components of the non-canonical pathway also depends on the PTM state of members of its degradation complex containing TRAF3/TRAF2/cIAP1/cIAP2 proteins, which keeps the level of NIK low under constitutive conditions. Upon activation, the degradation complex is recruited to the active receptor complex. This leads to the degradation of cIAP1-cIAP2, thus allowing NIK to dissociate from the complex and subsequently activate the non-canonical NF-κB pathway [73,74,75,76,77,78,79]. However, in an interesting finding in both normal B cells and B cell-derived tumors, it has been shown that CD40 or BAFF receptor activation results in the complete degradation of TRAF3 and partial degradation of TRAF2 but not cIAP1-cIAP2. These findings demonstrate a ubiquitination cascade in which TRAF2 ubiquitinates and activates cIAP1-cIAP2, which then ubiquitinates TRAF3, leading to its degradation and enhanced NIK stabilization as well as processing of NF-κB2/p100 [80]. Low levels of TRAF proteins lead to the higher accumulation of NIK, which then phosphorylates p100 and IKKα, thereby activating the kinase activity important for multiple site phosphorylation of p100 at its C-terminal. Phosphorylated p100 is ubiquitinated by β-TrCP, leading to cleavage-dependent ubiquitination-mediated degradation of C-terminal part of p100, generating active p52 subunit [56,81]. The processing of p100 is important in context to various steps of regulation in the activation of NF-κB pathway. Unprocessed p100 binds RelA, RelB or c-Rel subunit via its C-terminal ankyrin repeats, further inhibiting the activity of Rel subunits [82,83]. Hence, in the context of activation of the non-canonical NF-κB pathway, stabilization of NIK and processing of p100 acts as one of the major steps involving multiple PTMs of its regulator molecules [58].

Specific stimuli-dependent activation of the non-canonical NF-κB pathway is important in regulating various important biological functions such as lymphoid organogenesis, B-cell survival and maturation, dendritic cell activation and bone metabolism [84,85,86,87,88,89,90]. In spite of being tightly regulated by various activators and inhibitory factors, the aberrant activation of the NF-κB pathway has been observed in many lymphoid malignancies. Besides its role in immune regulation, NF-κB members have been documented to regulate transcriptional activities that promote the malignant transformation and survival of cancer cells. Several studies demonstrate the presence of promiscuous mutations responsible for inhibiting TRAF2, TRAF3 and cIAP1/2 complex or enhancing the expression/stability of NIK and other receptor molecules like CD-40 and LTβR. Such mutations are associated with the abnormal activation of the non-canonical NF-κB pathway [85,91,92,93,94,95]. Additionally, recent studies suggest the interdependency of NF-κB-driven expression of target genes with epigenetic changes in the genome. In this review, we will discuss the activation and regulation of NF-κB signalling in inflammation and cancer in context to its interaction with transcription factors (TFs), kinases, epigenetic modifiers and non-coding RNAs. We focus on discussing the interdependent role of NF-κB-signalling components with transcription factors and chromatin modifiers in the aberrant activation of the NF-κB pathway, as well as in the active transcriptional activation of its target genes (summarized in Figure 1).

## 2. NF-κB: One of the Key Factors Linking Inflammation in Cancer 

Even before Vichow’s hypothesis on the origin of cancer from the site of inflammation, several inflammation-associated viral and bacterial infections (Hepatitis B, *Helicobacter pyroli)* were found to be associated with increased risk of malignancies of the liver, colon and stomach [96,97,98,99]. Additionally, statistical reports estimate that inflammatory viral infections contribute to >15% of all cancers [100,101]. It has been postulated that cancer cells can hijack the normal inflammatory mechanism to boost their growth and survival. In general, the normal function of immune cells is to trigger the innate and adaptive immune response to differentiate between self/non-self and destroy/engulf the foreign invaders. However, in the tumour microenvironment, cancer cells can alter the protective functions of immune cells and convert them to act as tumour-promoting cells. They are reprogrammed to secrete pro-survival inflammatory cytokines, allowing better proliferation, survival, migration, invasion and inhibition of apoptosis in cancer cells. Hence, tumour-promoting inflammation acts as one of the major hallmarks of cancer. But what remains unanswered are the regulatory molecules that link inflammation to cancer progression. One of the major pathways reported to be involved in manipulating the immune response machinery in the tumour microenvironment is the NF-κB pathway. 

Several studies have converged on the role of the NF-κB pathway as one of the critical missing links between inflammation and cancer. The first evidence comes from various studies reporting extensive sequence similarity between c-Rel and viral oncoprotein v-Rel in their N-terminal domain, a region referred to as the Rel Homology Region (RHR), and identification of oncogene *Bcl3* as a member of the IκB family [102,103]. In addition, many cancer cell types show elevated levels of NF-κB expression and activation. Endogenous activation of NF-κB is reported in Hs294T melanoma cells due to altered equilibrium between IκBα degradation and resynthesis, leading to overall decrease in the level of IκBα expression [104]. Moreover, human colorectal cancer (CRC) epithelial cells have been observed to express enhanced NF-κB and IκBα, which is accompanied by the increased expression of *cox-2* gene [105]. Similar selective activation of the NF-κB pathway is also reported in breast cancer. The RelA subunit of NF-κB is reported to be activated in breast cancer cell lines, whereas breast tumours are shown to exhibit an absence or low level of nuclear RelA, in contrast to activated c-Rel, NF-κB1 and NF-κB2 along with *bcl2* expression, as compared to nontumorigenic adjacent tissue [106]. Most interestingly, the NF-κB family of transcription factors have been shown to contribute to the function and maintenance of tumour-initiating cells (TICs) in breast cancer. Experimental data indicates the activation of both the canonical and non-canonical NF-κB pathway to be important in the function of TICs by stimulating epithelial-to-mesenchymal transition (EMT) and upregulating the expression of the inflammatory cytokines IL-1β and IL-6 [107]. In a recent finding, Monica et al. reported the involvement of the NF-κB pathway towards resistance to endocrine and chemotherapies in breast cancer [108]. Furthermore, BRCA1 signalling, which is one of the prominent pathways activated in various cancer types including breast cancer and ovarian cancer, additionally possesses the capability to induce the NF-κB pathway [109]. 

Additionally, in lung carcinoma, the enhanced expression of IKKβ and NF-κB is reported as an important factor for tumour initiation and progression [110]. Another NF-κB activity dependent type of cancer is melanoma. Studies using mice models have shown that initiation of such tumours is HRas-mediated and involves the regulation by IKKβ in the activation of NF-κB [111]. In the cell line model of Diffuse large B-cell lymphoma (DLBCL), constitutive activity of IKK and high NF-κB DNA-binding activity is reported in the ABC-DLBCL subtype but not in the GC-DLBCL subtype [112]. Recently, Eluard et al. reported the presence of a new subset of Diffuse large B-cell lymphoma (DLBCL) patients showing enhanced RelB activation with aberrant gene expression and mutation profiles [113]. Based on these studies, it can be postulated that abnormal activation of the NF-κB pathway in various cancer types is critical for the survival of transformed cells, particularly in the suppression of apoptosis and senescence. Apart from endogenous elevated levels of NF-κB expression and activation in cancers (summarized in Table 1), there are also reports on aberrant activation of NF-κB pathway in cancer cells. Hence, there appears to be various extrinsic and intrinsic factors regulating the malignancy-associated enhanced activation of the NF-κB pathway in cancers. 

## 3. Factors Contributing to the Hyperactivation of NF-κB Pathway in Cancers

### 3.1. Cancer Associated Immune Signalling Molecules

In addition to the role of elevated NF-κB activity in the survival of transformed cells, NF-κB is found to be activated in cancer stem cells (CSCs). In CSCs, it promotes the release of pro-inflammatory cytokines that exert anti-apoptotic and pro-proliferative activities [129]. One of the prominent factors involved in the activation of the canonical NF-κB pathway in both solid and hematologic malignancies is the tumour microenvironment (TME). A large number of immune cells (macrophages, dendritic cells, neutrophils, mast cells, T cells and B cells) are recruited to the TME, leading to the enhanced production of cytokines, growth and angiogenic factors and proteases that degrade the extracellular matrix to support cancer development and progression [130]. In solid tumours, the sustained activation of the NF-κB pathway is predominantly achieved through the continuous release of cytokines by tumour-associated macrophages (TAM) in the TME. One of the predominant properties of TAM is the ability to switch from M1- to M2-phenotype with enhanced release of anti-inflammatory cytokines [131,132], suggesting a crosstalk between cancer cells and neighboring macrophages. Interestingly, IKKβ and NF-κB are also reported to assist in the polarization of macrophages towards the M2 type, which fosters and protects the tumour cells instead of attacking them [133,134]. Hence, this permits malignant cells to bypass tumour immunosurveillance activity via NF-κB-mediated polarization of macrophages from the M1 to M2 phenotype. 

Activation of either of the canonical or non-canonical pathways in both solid and hematologic malignancies also depends on different sets of inflammation-associated cytokines and receptors activated in tumour cells. Induction of the canonical NF-κB pathway is initiated by pattern recognition receptors and diverse tumour-promoting cytokines, such as TNF, IL-1, and IL-17 [135]. On the contrary, activation of the non-canonical NF-κB pathway is triggered by signalling via a specific subset of TNFR superfamily members such as B-cell-activating factor belonging to TNF family receptor (BAFFR) [52,55], CD-40 [53], lymphotoxin β-receptor (LTβR) [54], receptor activator for nuclear factor κB (RANKL) [136], TNFR2 [137,138], Fn14 [139], etc.

In addition to the involvement of immune regulatory molecules in activation of both the canonical and non-canonical NF-κB pathways towards cancer-promoting mechanisms rather than their classical immunosurveillance roles, the important observation is their expression in non-immune cells. Enhanced expression of CD-40 is reported in many non-immune cells, such as the intestinal epithelial cells (IECs) of patients with colon cancer. This, in turn, leads to the aberrant activation of non–canonical NF-κB pathway, suggesting the important link between immunosurveillance and tumorigenicity [140,141,142]. LTβR is expressed in lymphoid stromal and epithelial cells. BAFFR is predominantly expressed in B cells, whereas RANK, which is best known for its role in osteoclastogenesis, is also reported to be highly expressed in various cancer types like breast and prostate cancer cells, mediating the migration and skeletal metastasis of cancer cells [143,144]. 

### 3.2. Intrinsic Mutations and Post Translational Modifications (PTMs)

In addition to the receptor dependent aberrant activation of NF-κB pathways in cancer, activating mutations in other signalling components of the non-canonical NF-κB pathway have been documented particularly in lymphoid malignancies [91]. Such activation is driven by the presence of selected mutations inactivating the genes encoding negative regulators of the pathway (TRAF2, TRAF3, TNFAIP3, BIRC3, MAP3K14, CYLD, cIAP1/cIAP2) and activating the regulator molecules (NF-κB1, NF-κB2, CD40, LTβR, TACI, and NIK) in various cancer types like multiple myeloma (MM), splenic marginal zone lymphoma (SMZL), MALT lymphoma and B-cell lymphoma [15,93,145,146,147,148,149] (Table 2). Mutations leading to constitutive activation of the kinase NIK in multiple myeloma have been found in NIK itself, that disrupts its binding with TRAF3, in turn causing dissociation of NIK from the inhibitory complex having TRAF2 and the ubiquitin ligases cIAP1 and cIAP2. This, in turn, results in NIK stabilization, leading to aberrant activation of the non-canonical NF-κB pathway [93,119]. The genetic selection of these driver mutations by cancer cells highlights the critical importance of the NF-κB pathway towards cancer progression and enhanced malignancy. In the case of multiple myeloma, mutations are also reported in many other signalling subunits of non-canonical NF-κB pathway—NF-κB2, TRAF2, TRAF3, BTRC encoding β-TrCP, which alters the inhibitory degradative pathway of NIK kinase by TRAF2/TRAF3 complex, leading to malignancy-associated activation of non-canonical NF-κB signalling [150]. Overexpression of NIK due to t(17;22) chromosomal translocation is also associated with the occurrence of multiple myeloma [93]. Oncogenic mutations in the TP53 protein are reported to be associated with higher RelA expression, in turn activating the canonical NF-κB pathway in human B-cell lymphomas such as Hodgkin lymphoma and, to a lesser extent in T-cell lymphoma cell lines as well [151,152]. Chromosomal translocation t(10;14)(q24;q32) of the *NF-κB2* gene is observed to be associated with a variety of hematological malignancies, such as MALT lymphomas [153]. The translocation moves the IgG promoter to a region upstream of the *bcl-10* gene, resulting in expression of a truncated bcl-10 protein, leading to activation of NF-κB [153]. Another reported translocation is t(11;18), which results in the generation of a chimeric protein, AP12-MALT1, which leads to NF-κB activation in B-cell lymphomas [154]. The modified NF-κB2 gene codes for the protein that lacks the ankyrin regulatory domain but still binds the kappa B sequence in vitro. Such rearrangement of NF-κB2 has been reported in both B-cell and T-cell lymphoma patients suggesting that translocation dependent truncation of the ankyrin domain may be a common mechanism in the abnormal activation of *NFKB2* gene and its relevant role in lymphomagenesis [15,148,153,155,156,157]. Another genomic rearrangement event reported in DLBCL is on chromosome 10q24, which results in increased *NFKB2* mRNA expression, causing constitutive expression of NF-κB2 [15,158]. Chromosomal translocation of the *c-Rel* gene to chromosome 2p 13–15 causing its enhanced amplification has been reported in DLBCLs with a large cell component, constituting approximately 50% of B-cell non-Hodgkin’s lymphomas [149,159,160,161]. This chromosomal aberration of *c-Rel* has also been found in primary mediastinal (thymic) B-cell lymphomas and follicular large cell lymphomas, and is reported to be associated with extra nodal presentation [120]. Another member of the NF-κB family, *RelA,* is mapped to be translocated to 11q13, a site where a number of genes involved in neoplastic development have already been mapped, suggesting a link between chromosomal translocation and the tumour-inducing role of RelA [162]. Activating mutations (translocation t(14;19)(q32;q13.1)) in another member of the NF-κB pathway, *Bcl3*, which is a proto-oncogene, have also been observed in B-cell leukaemia. The chromosomal translocation of *Bcl3* results in its enhanced expression in leukemic cells as compared to normal blood cells [163]. In addition to biallelic deletion and chromosomal translocation, several other mutations, including missense mutation, frameshift mutation and in frame deletion are also reported to inactivate the *TRAF3* gene, in turn inducing the activation of NF-κB pathway [91].

Besides the reported genomic abnormalities, there are several other factors that can influence the transcriptional activity of NF-κB pathway. Reports suggest that crosstalk with certain activating and inhibitory kinases such as Glycogen Synthase Kinase (GSK-3β), p38 and PI3K can either modulate the transcriptional activity of NF-κB or its upstream signalling pathways [164,165,166]. Kinases are reported to modulate the activity of NF-κB in glioma cell lines and pancreatic cancer cells through post-translational modification (PTM) of the NF-κB subunits (p65/p50) [166,167,168,169]. Further studies indicate that GSK-3β has no role in the nuclear accumulation of NF-κB, but instead alters the DNA-binding activity of NF-κB subunits by inducing hypermethylation of the target DNA [167,168,169,170,171]. Other kinases documented to regulate the NF-κB pathway include the Jun- N-terminal kinase (JNK) and p38 [172]. Though both the kinases can be induced by the same stimuli (TNFα) that activate NF-κB pathway, they have been found to display differential functions on NF-κB activity. p38 acts as a co-factor to modulate the transactivation machinery of NF-κB to regulate TNF-induced IL-6 gene expression, whereas a counteracting relationship occurs between JNK and NF-κB. NF-κB complexes downregulate the c-Jun amino-terminal kinase (JNK) cascade via upregulation of *gadd45β*/*myd118* gene expression. Gadd45β, in turn, targets MKK7/JNKK2, a specific and essential activator of JNK. Mechanistically, binding of gadd45β with MKK7 blocks the catalytic activity of the latter, causing inhibition of the JNK pathway [173,174,175,176]. Hence, the aberrant activation of NF-κB pathway depends on multiple factors including cell type, micro-environment, PTMs, enzymatic activity of regulatory molecules and chromosomal abnormalities.

**Table 2 cells-12-00788-t002:** Summarized table of various mutations/chromosomal alterations in components of NF-κB pathway affecting its expression and activity in different cancer types.

Factors/Regulators of NF-κB Pathway	Type of Mutation/Chromosomal Translocation	Cancer Type	Response	References
*TRAF3*	Bi-allelic deletion at 14q32	MM	Increased p52/p100 ratios	[91,177]
*TRAF2*	Bi-allelic deletion at 9q34	MM	Increased p52/p100 ratios	[91,177]
*CYLD*	Bi-allelic deletion at 16q12	MM	Increased p52/p100 ratios	[91,177]
*cIAP1/cIAP2*	Bi-allelic deletion at 11q22	MM	Increased p52/p100 ratios	[91,177]
*NIK*	t(17;22) translocation, IgH translocation or amplification	MM	Overexpression of *NIK*	[93]
*LTBR*	amplification of the entire 12p chromosome arm	MM	Activatory	[91]
*NF-κB2*	t(10;14)(q24;q32) t(10q24)	MALT Lymphoma, DLBCL	Activatory, Enhanced expression of *NF-κB2* gene and protein	[15,91,153]
*AP12-MALT1*	t(11;18)(q21;q21)	B-cell Lymphoma, MALT lymphoma	Activatory	[154,155]
*c-Rel*	t(2p 13-15)	DLBCL, B-cell lymphoma, Follicular large cell lymphoma	Enhanced amplification of *c-Rel* gene	[120,121,149,159,161]
*RelA*	(11q13) site with t(11;14)(q13;q32)	NHL, Diffuse large cell lymphoma, Squamous carcinoma of head and neck, Breast cancer	Activatory	[159,162]
*Bcl3*	t(14;19)(q32;q13.1)	B-cell leukaemia	Activatory	[163]

### 3.3. Epigenetic Modification in the Component(s) of NF-κB Pathway

Reports suggest the dependency of NF-κB components on various epigenetic factors for its activation in cancer cells. Reduced expression of histone methyltransferase EZH2 stimulates the expression of TRAF2/5 via the de-repression of their expression due to H3K27 hypermethylation by EZH2. The elevated TRAF2/5 expression, in turn, enhances TNFα-induced activation of NF-κB signalling, leading to an uncontrolled inflammatory reaction which ultimately contributes to tumorigenesis [178]. The enhanced activation and expression of NF-κB-signalling component proteins in various cancer types also depends on the epigenetically modified state of its own component genes and its target genes. Triple negative breast cancer cells display a high level of NF-κB activation due to the enhanced expression of NIK that is caused by the epigenetic alteration (histone H3 acetylation) of the *NIK* gene [179]. Hence, these studies suggest the plausible de-regulation of the NF-κB pathway due to epigenetic alterations.

## 4. Double Edged Role of NF-κB from Immunosurveillance to Pro-Tumorigenic Functions

As discussed, the aberrant activation of the NF-κB pathway in cancer is a multifactorial event. Depending on the prevalent tumour microenvironment, malignancy-promoting mutations in the components of the NF-κB-signalling cascade, and the inflammatory molecules released by the tumour immune cells, the biological importance of the NF-κB pathway is diverted from the immunosurveillance mechanism towards tumour-promoting functions.

NF-κB signalling has been shown to activate the expression of various inflammatory mediators, such as IL1β, TNF and IL6, which promote cancer development [180,181]. However, the question remains as—what factor(s) drives the variation in inflammatory response by the NF-κB pathway from a protective role towards a tumour-promoting role. The answer to this oncogenic shift is related to the severity of inflammation response which mostly occurs during chronic inflammatory conditions. During acute inflammatory conditions, NF-κB activation acts as a tumour immunosurveillance mechanism to assist in the targeting and elimination of transformed cells. For example, protein kinase D1-mediated activation of NF-κB signalling can induce the expression of antioxidant proteins such as MnSOD and anti-apoptotic proteins including A20 and cIAPs to prevent the accumulation of pro-tumorigenic ROS that can cause oncogenic mutations [182,183,184,185,186]. NF-κB-mediated inhibition of ROS accumulation can also repress the activity of pro-tumorigenic transcription factors such as STAT3 and AP1 [185]. In contrast, under chronic inflammatory conditions, the continuous presence of NF-κB stimuli seem to outperform the inhibitory role of the negative NF-κB regulators, leading to constitutive activation of NF-κB signalling. Such constitutive activity of NF-κB can exert pro-tumorigenic effects ranging from cell proliferation and cell survival to malignant cell invasion and metastasis. Many cancers arise from sites of chronic infection or inflammation due to elevated ROS production by neutrophils in response to invading pathogens. This innate immune response in turn causes DNA damage and genetic mutations, thereby triggering tumour initiation [186,187]. 

Though both the canonical and non-canonical NF-κB pathways are reported to be activated in various invasive and malignant cancers, the functional mechanism for downstream substrates involved in activation of the non-canonical NF-κB pathway are well characterized compared to the canonical pathway. The invasive nature of Glioblastoma Multiforme (GBM) cells has been reported to be associated with high RelB expression [116,117]. Work on mouse tumour xenograft models also showed activation of the non-canonical NF-κB pathway leading to regulation of the expression of its own regulator protein NIK, which, in turn, is reported to induce dramatic cell shape changes, increase tumour cell invasion and promote aggressive orthotopic tumour growth [123]. Point mutations at the promoter region of the *telomerase reverse transcriptase* (*TERT*) gene is one of the most frequent non-coding mutations in cancer. *TERT* promoter mutations (TPMs) are cancer type-specific and among the first few mutations reported in melanomas, glioblastomas and hepatocellular carcinomas [188,189,190,191]. In an interesting finding, non-canonical NF-κB signalling is reported to drive the expression of the *TERT* gene carrying −146 C > T mutation in its promoter region, causing telomerase reactivation, which is otherwise not activated via binding of ETS transcription factor [192,193]. This data specifically highlights a novel role of the non-canonical NF-κB pathway in the reactivation of telomerase in cancers. Hence, the level of inflammatory response and genetic changes in the cancer cells can act as some of the major factor(s) deciding the difference between acute inflammatory response versus aberrant activator response of the NF-κB-signalling pathway in cancer. 

## 5. Aberrant NF-κB Activation Driven Expression of Tumour Promoting Genes

Apart from activating the expression of its immune response target genes, aberrantly activated NF-κB signalling in cancer cells contribute to cancer progression by acting as a transcriptional activator of various other pro-tumorigenic genes involved in cell proliferation, inhibition of apoptosis, invasion, metastasis and angiogenesis. 

In-depth studies also show that NF-κB controlled genes regulating oncogenic properties are significantly different. NF-κB-dependent cancer-relevant genes mostly encode for cytokines, cell cycle genes like cyclin D1, matrix metalloproteinases (MMPs) and anti-apoptotic proteins. Numerous NF-κB target genes such as *cIAP1/2*, *TRAF1/2*, *Bcl-xL*, *XIAP*, *MnSOD* and *IEX-1L* confer antiapoptotic properties [106,194,195,196]. Specifically, the NF-κB target gene *cIAP1/2* functions as an inhibitory factor of cancer cell apoptosis through directly binding and suppressing the effector caspases [197,198]. NF-κB signalling controls the epithelial to mesenchymal transition and metastasis, often via upregulation of matrix metalloproteinases (MMPs) [199]. In breast cancer, NF-κB is also reported to induce the expression of EMT-related genes such as *Twist*, intercellular adhesion molecule-1 (*ICAM-1*), endothelial leukocyte adhesion molecule 1 (*ELAM-1*), vascular cell adhesion molecule 1 (*VCAM-1*), MMPs and serine protease urokinase-type plasminogen activator (uPA), along with the expression of one of the major tumour promoting genes *Bcl2* [200,201]. Interestingly, one study revealed a role for NIK in the phosphorylation, enzymatic activity and pseudopodal localization of membrane type 1 MMP in highly invasive tumours like glioblastoma that is distinct from its established kinase function in the non-canonical NF-κB pathway [202]. 

NF-κB signalling also contributes to tumour progression and invasion by controlling pro-angiogenic genes such as vascular endothelial growth factor (*VEGF*) and its receptors, macrophage inflammatory protein-1 (*MCP-1*) and CXC-chemokine ligand 8, also known as IL-8 (*CXCL8*) [203,204,205,206,207]. Activated NF-κB signalling in cancer transactivates the expression of *cyclin D1* and *c-myc* that promote cancer cell proliferation [208,209]. Angiogenesis, the phenomenon of new blood vessel formation is one of the hallmark phenotypes of cancer cells. Tumour angiogenesis is dependent on proinflammatory cytokines, chemokines and growth factors such as MCP-1, IL-8, TNF-α and VEGF, secreted by tumour-associated macrophages (TAMs) via the activated NF-κB pathway. Furthermore, the recruitment of bone marrow-derived cells (BMDCs) to tumours for vasculogenesis is essential for tumour angiogenesis, which is found to involve NF-κB-mediated enhanced expression of IL-8 and angiogenin [210,211]. Subsequently, the expression and activation level of different NF-κB subunits can induce varying severity in different cancer types. In the case of ER-positive breast carcinoma, higher expression of RelB is associated with decreased relapse-free survival (RFS) and overall survival (OS) rate, whereas in other tumours, such as lung carcinoma, enhanced expression of NIK and RelB is associated with enhanced metastasis and shorter OS. Poor RFS outcome is reported to be associated with higher expression of non-canonical NF-κB target gene myoglobin [212,213,214]. Elevated RelB activity reported in a new subset of DLBCL patients is found to confer resistance to DNA damage-induced apoptosis along with increased *cIAP2* expression [113]. In a more recent finding, sustained activation of the non-canonical NF-κB signalling is also shown to drive doxorubicin resistance in DLBCL via enhanced glycolysis [215]. Hence, these studies indicate the existence of a high degree of NF-κB dysregulation in cancer. 

## 6. Different Modes of Deregulated NF-κB Signalling in Cancer

While we discussed the multifaceted roles of the NF-κB pathway linking inflammation and cancer, it is also important to understand the interacting map of the components of this pathway with other macromolecules, which, in turn, regulate the transcription of pro-oncogenic transcripts (Figure 2).

### 6.1. Interaction with Transcription Factors

While NF-κB regulates the expression and activity of various regulatory factors, its own activity can also be regulated via direct association with several other transcription factors. The most prominent ones are proto-oncogenic transcription factors such as STAT3, p53, AP1 and ETS-related genes *ERG*, implicating their plausible cooperative function with NF-κB factors in inflammation and cancer [216,217,218]. Hence, depending on the promoter sequence and structure of the target genes, the functional link between NF-κB and other transcription factors might vary. One of the well-characterized factors known to co-associate with NF-κB is the STAT family members. NF-κB, in association with STAT3, regulates the expression of various cell cycle genes, anti-apoptotic genes and genes encoding cytokines and chemokines [219]. Studies suggest that the direct interaction of RelA and NF-κB1 members with STAT3 facilitates both the recruitment of NF-κB and STAT3 onto each other’s promoter sites [220,221,222]. In another context of regulation, STAT3 modifies the RelA subunit by recruiting acetyltransferase p300, resulting in the acetylation-dependent retention of NF-κB in the nucleus [223]. Such regulation leads to the enhanced activity of NF-κB (a tumour-promoting phenomenon) and hence, chronic stimulation of cytokines in the tumour microenvironment. Cross talk of NF-κB with transcription factor p53 also occurs [221]. Enhanced secretion of the pro-inflammatory cytokine TNFα triggers the formation of an active complex containing nuclear RelA and p53 on κB binding motifs, suggesting the importance of p53 in NF-κB-mediated gene expression induced by canonical stimuli [224,225]. In addition, some reports suggest that the RelA subunit and transcription factor p53 can regulate their respective transcriptional activities. p53 has been shown to inhibit NF-κB transcriptional activity, while the RelA subunit can also inhibit p53-dependent transactivation of target genes [221]. This constitutive activation of NF-κB, evoked by a p53 hot-spot mutant protein frequently found in tumours, provides an explanation for the fact that p53 mutations arise more than p53 deletions in tumours of various origin [222,226]. More recently, another transcription factor, the ETS family member ERG, has been identified to cross talk with NF-κB. As reported by various groups, the functional role of ERG is validated in various leukemia, Ewing sarcoma and prostate cancer [227,228,229,230]. Interestingly, NF-κB activation is elevated in ERG fusion-positive prostate cancer patients and cancer cell lines [231]. ERG is also reported to regulate expression of the NF-κB target gene, *ICAM-1* in endothelial cells [232,233]. Another interesting study also revealed the cooperative function of p52 with transcription factor ETS1 in the reactivation of telomerase in cancers via a hotspot −146 C > T *TERT* promoter mutation [192]. On a similar line, a recent finding has shown the involvement of the non-canonical NF-κB pathway in altering the genomic binding landscape of transcription factor ETS1 that supports glioma progression [234]. Hence, a cross talk is predicted between NF-κB and other TFs at the level of activation and transcriptional regulation of NF-κB target genes, which requires further studies for in depth understanding of the mechanisms involved. 

### 6.2. Effect of Pro Tumorigenic Non-Coding RNAs

Considering the challenge with highly evolving cancer cells which are resistant to many available therapies either through selected genetic mutations or positive adaptation to the cancer microenvironment, it is critical to understand new alternative modes of regulations adopted by cancer cells. In recent times, one such regulatory molecule showing relevance in context to its crosstalk with the components of NF-κB pathway is non-coding RNAs (ncRNAs). Altered regulation at the level of epigenome mediated by non-coding RNAs (microRNAs—miRNAs and long noncoding RNAs—lncRNAs) has been found to be a prevailing factor impacting various types of malignancies. Several miRNAs are transcriptional targets of NF-κB, such as *miR-9*, *miR-21*, *miR-143*, *miR-146* and *miR-224*, which, in turn, act as a feedback mechanism for modulating the activity of NF-κB [235,236,237,238,239,240,241]. Out of these, *miR-21* and *miR-143* are reported to be involved in regulating the malignant phenotypes like invasion and metastasis in cancer types including breast cancer and HCC [238,239]. On the other hand, NF-κB can also induce the expression of proteins important for the transcriptional regulation of miRNAs. One such example is the NF-κB driven expression of Lin28 protein, which inhibits the processing and maturation of *let-7* miRNAs—a family of tumour suppressor miRNAs whose expression is downregulated in many cancer types. *Let7* miRNA also targets *IL6*. Thus, Lin28-mediated downregulation of *Let7* miRNA leads to the higher expression of *IL6* and further enhances NF-κB signalling in a positive feedback loop mechanism [242]. 

Subsequently, NF-κB activity is also regulated by the presence of several miRNAs mostly via repressive mechanisms. One such miRNA is *miR-502e*, which is reported to act as a tumour suppressor factor by altering cell proliferation in hepatoma cell lines and hepatocellular carcinoma by targeting NIK, thereby modulating the activity of non-canonical NF-κB signalling [243]. Many highly expressed long non-coding RNAs (lncRNAs) are also reported to regulate the activity of NF-κB. The lncRNA *NKILA*, was reported to mask the phosphorylation motifs of IκB, further inhibiting the activation of NF-κB [244]. Along with the aberrant activation of the NF-κB-signalling pathway, the expression of long non-coding RNAs (ncRNAs) is also dysregulated in different types of cancer cells, further regulating the degree of malignancy. The upregulated expression of lncRNA *H19* in melanoma cells and *Helicobacter pylori*-induced expression of *H19* in gastric cancer cells have been reported to be associated with enhanced cancer cell invasion and migration via activation of the NF-κB- and PI3K/Akt-signalling pathways [245,246]. Another NF-κB-associated lncRNA reported to be upregulated in cancer cells is lncRNA *NEAT1*. Its overexpression promotes proliferation, migration and invasion, influences the expression of EMT markers, and activates the NF-κB pathway in HeLa and SiHa cells [247]. *H19* and *NEAT1* are also reported to be associated with the resistance of cancer cells to chemotherapeutic drugs including bortezomib and dexamethasone respectively [248,249]. Hence, it can be speculated that subunits of NF-κB function in association with ncRNAs to impart their pro-tumorigenic roles along with chemoresistance functions in tumour cells whose mechanism remains elusive and requires further clarification.

## 7. Role of NF-κB Signalling in Shaping the Cancer Cell Chromatin Landscape 

Though the NF-κB family of proteins lack endogenous chromatin modifying enzymatic activity, they can exert changes in the chromatin landscape either by acting as a mediator to recruit and position chromatin modifiers onto target genes in a specific sequence dependent manner or by regulating the expression and activity of those modifiers [63,250]. One noteworthy feature of NF-κB family members is their ability to form multimeric complexes. Apart from forming multimeric complexes with its own family proteins, NF-κB subunits are reported to form complexes with other proteins, which includes chromatin modifiers as well. Upon lymphotoxin treatment, non-canonical NF-κB signalling is activated and RelB/p52 dimer gets associated with the SWI/SNF chromatin remodeling complex via an adapter protein, requiem, to induce the expression of the *BLC* gene (*CXCL13*). Such interaction suggests an indirect role of activated NF-κB signalling in the epigenetic regulation of oncogene expression [250]. Additionally, the NF-κB pathway also acts as a key regulator in the enhanced expression of chromatin modifiers and its subunits/interacting proteins, such as Enhancer of Zeste Homologue 2 (EZH2), a histone-lysine N-methyltransferase enzyme involved in the epigenetic modification of histone protein (H3K27), thus conferring the hypermethylation-mediated repressive gene expression of anti-oncogenic genes [251]. In colorectal cancers, NF-κB activation in response to TNFα has been reported to induce the expression of EZH2, leading to the inhibitory promoter hyper-methylation of pro-apoptotic protein kinase cδ binding protein (PRKCDBP) and resultant increased growth of cancer cells [252]. Subunits of the NF-κB pathway can also act in a de-repression mechanism to remove the repressive chromatin marks and complexes. Some inducible gene promoters harbor high levels of the H3K9 dimethyl modification, associated with transcriptional silencing. However, upon stimulation, these marks are removed by the Aof1 histone demethylase, whose recruitment requires initially bound c-Rel dimers within the promoter region [253,254]. The NF-κB pathway is also reported to regulate RNA Polymerase II elongation by changing the chromatin landscape via recruitment of General Control Non-Derepressible 5 (GCN5) acetyltransferase complexes that primarily modify H4K5/K8/K12 lysine residues. The accumulation of acetylated H4 histone proteins leads to the association with BRD4, which then positively regulates transcription by recruiting the elongation factor P-TEFb [255]. Hence, the ability of the components of the NF-κB pathway to alter the chromatin landscape is not only limited to its signature DNA binding property but also extended to the recruitment of various chromatin modifiers assisting in transcriptional regulation. 

## 8. Concluding Remarks

Since the discovery of NF-κB nearly four decades ago, the multi-faceted roles of NF-κB members and their new transcription-binding partners in cancer have been gaining more clinical relevance in recent years. Although inflammation was previously implicated to promote the malignancy of human cancers, the causal mechanisms underscoring the link between inflammation and cancer have not been adequately characterized. Recent studies showing the aberrant activation of the NF-κB pathway in various cancer types and the regulation of NF-κB members in various tumorigenic events support the role of NF-κB as a hub linking inflammation and cancer. Although the occurrence of activating mutations in the NF-κB pathway is predominantly observed in hematological malignancies, the activation of NF-κB in solid tumours is also not negligible. The functional shift of the NF-κB pathway from inflammation to oncogenesis is mostly driven by the onset of chronic inflammatory conditions. NF-κB members can exert pro-oncogenic functions during cancer development through the activation of target gene transcription by their heterodimers. In addition, NF-κB components have also been demonstrated to interact with other factors, including transcription factors, kinases, epigenetic modifiers and other biological molecules like ROS and ncRNAs, to drive multiple oncogenic activities. Despite substantial progress in the understanding of various aspects of NF-κB signalling in cancer, the approaches for the targeted inhibition of specific components in the signalling pathway are limited due to various challenges. These challenges arise from the complex nature of its activity in different cancer types. Recent genomics studies have revealed the active selection of a wide range of driver mutations in cancer cells, some of which are important to facilitate the activation of the NF-κB pathway. In addition, epigenetic alterations have been documented to contribute to the aberrant activation of the NF-κB pathway. Conversely, the activated NF-κB pathway is also reported to confer changes in the chromatin landscape of cancer cells towards enhanced malignant phenotypes. Hence, these findings can potentially pave new ways for the development of precision medicine to improve the efficiency of existing cancer therapies and overcome the phenomenon of multidrug resistance in most of the cancer types.

## Figures and Tables

**Figure 1 cells-12-00788-f001:**
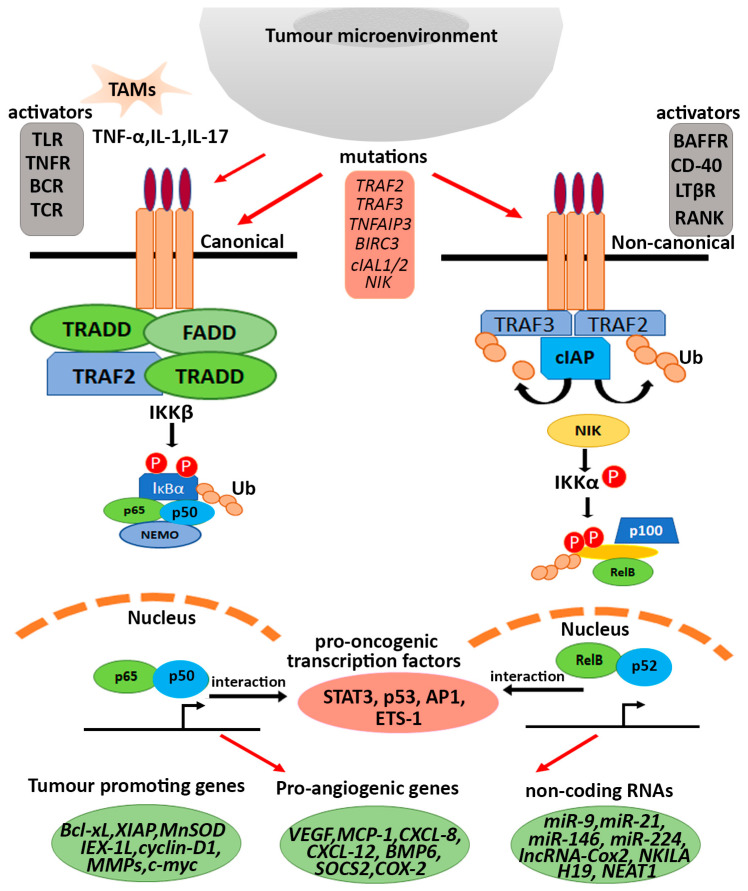
Aberrant activation and transcriptional regulation of NF-κB pathway in cancer. In addition to constitutive activator molecules of both the canonical and non-canonical NF-κB pathway (highlighted in grey box), activation of the NF-κB pathway in cancer occurs via various other factors involving cytokines like TNF-α, IL-1, IL-17 secreted by tumour-associated macrophages (TAMs) and oncogenic driver mutations in various regulatory factors of the pathway (highlighted in pink box). Upon such activation, the activated NF-κB subunits interact with pro-tumorigenic transcription factors (highlighted in red oval), causing activation of alternative target genes associated with tumour promotion phenotypes, angiogenesis and tumour-associated non-coding RNAs.

**Figure 2 cells-12-00788-f002:**
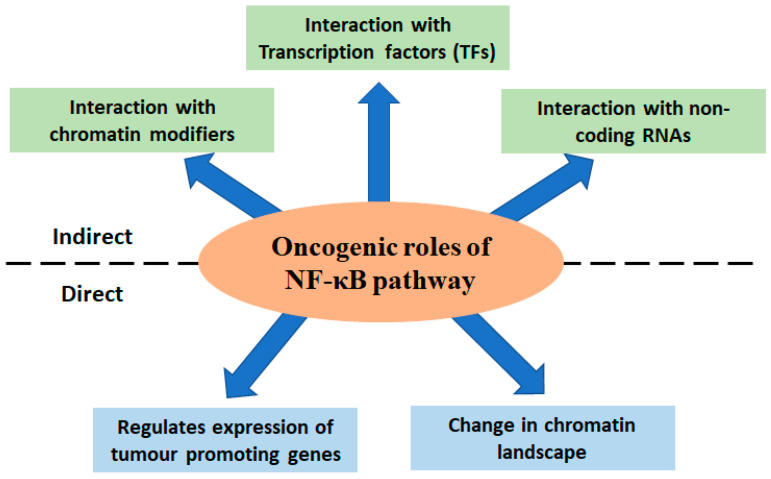
Graphical summary of the mode of activation of oncogenic pathways by NF-κB signalling cascade via direct (bottom panel) and indirect (upper panel) interaction/regulations of various macromolecular components and factors.

**Table 1 cells-12-00788-t001:** Summarized table of alteration of different components of NF-κB pathway in different cancer types (HCC: Human Colorectal Carcinoma; FLHCC: Fibrolamellar Hepatocellular Carcinoma; GBM: Glioblastoma Multiforme; MM: Multiple Myeloma; DLBCL: Diffuse large B-cell lymphoma; PMBL: Primary Mediastinal B-cell Lymphoma; ABC-DLBCL: Activated B-cell-like Diffuse large B-cell Lymphoma; PEL: Primary Effusion Lymphoma; ALT: Adult T-cell Lymphoma/Leukemia).

Factors of NF-κB Pathway and Alterations	Cancer Type	References
RelA (p65) Higher expression and activation of p65 in tumour tissue compared to normal tissue	HCC, FLHCC, Breast cancer cell lines	[105,114,115]
RelB Frequent activation of RelB leading to non-canonical NF-κB pathway activation	GBM, MM, DLBCL	[113,116,117,118,119]
c-Rel Gain of chromosomal material (9p) leading enhanced expression of *c-Rel* Gain of 2p leading to nuclear accumulation of c-Rel	Breast tumours, Hodgkin lymphoma, DLBCL, GC-DLBCL, PMBL	[120,121,122]
NF-κB–inducing kinase (NIK) Enhanced activation/overexpression of NIK in cancer cells/tumour compared to normal	GBM, MM, Hodgkin lymphoma	[110,119,123]
IκBα Enhanced degradation of IκBα causing endogenous activation of NF-κB pathway High expression of IκBα transcript in certain cancer type Direct transactivation via interaction with κB motifs in the DNA inside the nucleus	Ovarian Carcinoma cell line, HCC, Melanoma	[104,105,109,124,125]
IKK Higher expression of endogenous IKK in tumour tissue compared to normal tissue Constitutive IKK activity causing enhanced activation of NF-κB pathway	Lung carcinoma, Melanoma, Hodgkin lymphoma, ABC-DLBCL, PEL, ALT, HCC	[105,110,111,112,126,127,128]
NF-κB1 Higher expression of p50 in tumour tissue compared to normal tissue Enhanced degradation of IκBα leads to increased activation of NF-κB1 subunit	Breast tumours, Melanoma cells, Lung cancer, HCC, ALT, ABC-DLBCL	[104,105,110]
NF-κB2 Selective activation in some cancer types Overexpression of NIK underlies constitutive activation of non canonical NF-κB pathway	Colon carcinoma cell line, MM, Breast cancer, Lung cancer	[106,107,110,119,124]

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
