# Peer review of "Transcriptional Regulation during Aberrant Activation of NF-κB Signalling in Cancer"

_cells, 2023, doi:10.3390/cells12050788_

Round 1
Reviewer 1 Report
The manuscript submitted by Deka and Li discusses the role of NK-kB in regulating signaling pathways in cancers. The manuscript is well-written and is a comprehensive review of the literature covering topics regulated by NF-kB. Below are a few suggestions to improve the draft:
1. In the introduction section, the authors discuss how the phosphorylation of the p100 subunit leads to the ubiquitination and processing of p100 to the active p52 subunit. But there are inconsistencies if this processing is “proteasome-mediated” (as stated in line 49) or “cleavage dependent” (lines 105-106). It would be good to clarify this for the readers.
2. Figure 2. The red color for the driver mutation and pro-tumorigenic transcription does not appear red in the submitted draft. Consider changing the color and amending the figure legend accordingly.
3. The name of the genes should be in italics.
4. There are multiple typological errors. Please proofread and modify the manuscript.
Author Response
Response: The authors highly appreciate the time and efforts given by the esteemed reviewers to go through our review in depth and providing us with valuable comments to improve our writing as well as the content of the review for wider benefit of the readers.
Considering the comments, we tried our best to incorporate all the changes required along with addition of one more detailed table and few more new sections. The detailed changes done along with line number is provided in the table below for reference and recheck.
Sl.No |
Comments |
Response |
Line number (in edited copy) |
1 |
In the introduction section, the authors discuss how the phosphorylation of the p100 subunit leads to the ubiquitination and processing of p100 to the active p52 subunit. But there are inconsistencies if this processing is “proteasome-mediated” (as stated in line 49) or “cleavage dependent” (lines 105-106). It would be good to clarify this for the readers |
It has been changed to “cleavage and ubiquitination mediated degradation of C-terminal half of p100 protein generating active p52 subunit”. |
Line 52-53 Line 137-138 |
2 |
Figure 2. The red color for the driver mutation and pro-tumorigenic transcription does not appear red in the submitted draft. Consider changing the color and amending the figure legend accordingly |
The legend has been changed according to the color (pink). |
Line 167 |
3 |
The name of the genes should be in italics. |
All the names of the gene are changed to italics (even in Figure 1). |
|
4 |
There are multiple typological errors. Please proofread and modify the manuscript. |
Authors appreciate the thorough revision from the reviewer. We tried our best to proofread and have corrected the visible typological errors to the maximum. |
|
Reviewer 2 Report
The review article “Transcriptional regulation during aberrant activation of NF-κB signalling in cancer” by Deka et al. provides an informative description of NF-κB as a hub linking inflammation and cancer. They first described the NF-κB family and its signaling pathways. They then elaborated on NF-κB’s role in linking inflammation to cancer progression; and its role in both immunosurveillance and pro-tumorigenic functions. They also discussed NF-κB’s interactions with other transcription factors and non-coding RNAs. Finally, they attempt to chart the link between NF-κB signaling and chromatin landscape of cancer cells towards enhanced malignant phenotypes. In sum, this is a very interesting and informative review; however, the text would be benefit from further editing to improve clarity.
Please find below the specific comments.
1. Inconsistent writing of NF-κB: “NF-κB”, “NFκB”, “NF- κB”(a space was added after the hyphen). Please be consistent.
2. There are canonical and non-canonical NF-κB signaling pathways, also know as classical and alternative pathways. Again, please be consistent through out the writing, don’t switch names.
3. Different fonts were used within a figure which looks un-professional; and p50, p52 should both should start with small letter p for protein name.
4. Lines 31-32, “The family of NF-κB transcription factors comprises of five members – RelA (p65), RelB, c-Rel, NF-κB1 (p105/p50) and NF-κB2 (p100/p52) [4-8].”
a. Please be specify this is the mammalian family;
b. Some of the cited references are the proper ones should be cited; suggest the authors to cite the original research articles in discovering all NF-κB family members.
5. Line 37, “The IκB family consists of three members (IκBα, IκBβ, and IκBε), all sharing similar structures.”. There are more than three members in the IκB family, please add other IκB proteins.
6. Lines 43-44, “Upon activation, IκB kinases phosphorylates inhibitory IκBs…”. Please specify which signalling pathways this is.
7. Lines 46-47, “the presence of nuclear localised IκB proteins that are capable of terminating NF-κB dependent gene transcription inside the nucleus”. Not all nuclear IκB proteins terminate the NF-kB transcription; for example, IκBbeta has been shown to bound to NF-κB on target gene promoter and etc. Please elaborate the discussion.
8. Lines 73-76, “RelA (p65), RelB and c-Rel contain the carboxy-terminal transactivation domains (TAD), which forms transcriptionally active heterodimers only with p50 and p52 subunits, in turn assisting in DNA binding activity and activated target gene expression. [31, 32].” RelA, RelB and cRel not only form heterodimers wit p50 and p52, there are also RelA:RelA and c-Rel:c-Rel homodimers.
9. Lines 108-109, “Unprocessed p100 binds RelB and RelA subunit via its C-terminal ankyrin repeats, further inhibiting the activity of Rel subunits”. Proper reference(s) should be added.
10. Lines 175-177, “In a recent finding, Monica et al., also reported the involvement of NF-κB pathway towards resistance to endocrine and chemotherapies.” Reference should be added.
11. It would be useful if in Table 1, where the different types of alteration should be listed and associates with cancer types.
12. Lines 232-233, “suggesting the important link between immunosurveillance and tumorigenicity. [101-103].” Please delete the full-stop before references.
13. In the section “3.2. Intrinsic mutations and post translational modifications (PTMs)”, there is a long paragraph with detailed description of mutations/chromosomal translocation in varies NF-κB genes which enhance cancer progression and malignancy.
a. I think it could be useful to add a table to summarize which will be easier to read;
b. Specific driver mutations/PTMs should also be discussed in this section; and adding a table listing mutations, also listed which cancer types these come from will be useful as well.
14. Line 488, “Iκ-B” the hyphen should be deleted.
15. Lines 504-507, “Though NF-κB family of proteins lack endogenous chromatin modifying enzymatic activity, they can exert changes in the chromatin landscape either by acting as a mediator to recruit and position chromatin modifiers onto target genes in a specific sequence de-pendent manner or by regulating the expression and activity of those modifiers [211].”. [211] is not the proper reference.
16. Line 538, “Since the discovery of NF-κB more than two decades ago…” Since its discovery in 1986, it has been more than three decades, nearly four decades by now.
Author Response
Response: The authors highly appreciate the time and efforts given by the esteemed reviewers to go through our review in depth and providing us with valuable comments to improve our writing as well as the content of the review for wider benefit of the readers.
Considering the comments, we tried our best to incorporate all the changes required along with addition of one more detailed table and few more new sections. The detailed changes done along with line number is provided in the table below for reference and recheck.
Sl.No |
Comments |
Response |
Line number (in edited copy) |
1 |
Inconsistent writing of NF-κB: “NF-κB”, “NFκB”, “NF- κB”(a space was added after the hyphen). Please be consistent. |
We appreciate the thorough review of our manuscript by the reviewer. Throughout the review, we have corrected NF-κB as “NF-κB”. |
|
2 |
There are canonical and non-canonical NF-κB signaling pathways, also known as classical and alternative pathways. Again, please be consistent throughout the writing, don’t switch names. |
We have changed the nomenclature in the entire review as follows: Classical pathway: named as “canonical” Alternative pathway: named as “non-canonical pathway”. |
|
3 |
Different fonts were used within a figure which looks un-professional; and p50, p52 should both should start with small letter p for protein name |
Font size and font style have been made uniform for Figure 1. Also, we have changed p50, p52, p65 in the figure. |
|
4 |
Lines 31-32, “The family of NF-κB transcription factors comprises of five members – RelA (p65), RelB, c-Rel, NF-κB1 (p105/p50) and NF-κB2 (p100/p52) [4-8].” · Please be specify this is the mammalian family · Some of the cited references are the proper ones should be cited; suggest the authors to cite the original research articles in discovering all NF-κB family members |
· Specified as mammalian · Original research findings are being cited now. |
Line 31-32 |
5 |
Line 37, “The IκB family consists of three members (IκBα, IκBβ, and IκBε), all sharing similar structures.”. There are more than three members in the IκB family, please add other IκB proteins. Added |
It has been corrected and all the member proteins of IκB family have been added in the text with additional information. |
Line 37-42 |
6 |
Lines 43-44, “Upon activation, IκB kinases phosphorylates inhibitory IκBs…”. Please specify which signalling pathways this is |
The pathway has been specified “canonical NF-κB pathway”. |
Line 48 |
7 |
Lines 46-47, “the presence of nuclear localised IκB proteins that are capable of terminating NF-κB dependent gene transcription inside the nucleus”. Not all nuclear IκB proteins terminate the NF-kB transcription; for example, IκBbeta has been shown to bound to NF-κB on target gene promoter and etc. Please elaborate the discussion |
The functional role of nuclear localised IκB member proteins has been elaborately written in the revised review. |
Line 53-77 |
8 |
Lines 73-76, “RelA (p65), RelB and c-Rel contain the carboxy-terminal transactivation domains (TAD), which forms transcriptionally active heterodimers only with p50 and p52 subunits, in turn assisting in DNA binding activity and activated target gene expression. [31, 32].” RelA, RelB and cRel not only form heterodimers wit p50 and p52, there are also RelA:RelA and c-Rel:c-Rel homodimers. – |
Studies on homodimerization of p65, p50, p52 along with Rel proteins have been added in subsequent section in the revised review. |
Line 103-113 |
9 |
Lines 108-109, “Unprocessed p100 binds RelB and RelA subunit via its C-terminal ankyrin repeats, further inhibiting the activity of Rel subunits”. Proper reference(s) should be added |
New references are added. |
Line 141 |
10 |
Lines 175-177, “In a recent finding, Monica et al., also reported the involvement of NF-κB pathway towards resistance to endocrine and chemotherapies.” Reference should be added |
Reference has been added. |
Line 209 |
11 |
It would be useful if in Table 1, where the different types of alteration should be listed and associates with cancer types |
The table has been modified with addition of different alterations reported in NF-κB factors. For mutation and chromosomal alterations, we added a new Table 2. |
|
12 |
Lines 232-233, “suggesting the important link between immunosurveillance and tumorigenicity. [101-103].” Please delete the full-stop before references. |
This has been corrected. |
Line 269 |
13
|
In the section “3.2. Intrinsic mutations and post translational modifications (PTMs)”, there is a long paragraph with detailed description of mutations/chromosomal translocation in varies NF-κB genes which enhance cancer progression and malignancy. · I think it could be useful to add a table to summarize which will be easier to read; · Specific driver mutations/PTMs should also be discussed in this section; and adding a table listing mutations, also listed which cancer types these come from will be useful as well14
|
A table has been added here (Table 2). |
Line 346 |
14 |
Line 488, “Iκ-B” the hyphen should be deleted |
I-κB has been renamed as “IκB” throughout the review. |
|
15 |
Lines 504-507, “Though NF-κB family of proteins lack endogenous chromatin modifying enzymatic activity, they can exert changes in the chromatin landscape either by acting as a mediator to recruit and position chromatin modifiers onto target genes in a specific sequence de-pendent manner or by regulating the expression and activity of those modifiers [211].”. [211] is not the proper reference. |
Relevant refence has been added. |
Line 547 |
16 |
Line 538, “Since the discovery of NF-κB more than two decades ago…” Since its discovery in 1986, it has been more than three decades, nearly four decades by now |
This has been changed to “nearly four decades”. |
Line 578 |
Reviewer 3 Report
The work you presented in showing regarding the role of NF-kB in cancer shows that role is in-fact playing major role in cancer proliferation and regulation with immune cells. This would be really impactful for the readers and the biotech company who are working towards multiple different cancer targets. This detail paper will help understand how to use this pathway and study in detail to find relevant therapy using different modality.
Author Response
Response: The authors highly appreciate the time and efforts given by the esteemed reviewers to go through our review in depth and providing us with valuable comments and words of appreciation.